# Molecular Docking Study on Several Benzoic Acid Derivatives against SARS-CoV-2

**DOI:** 10.3390/molecules25245828

**Published:** 2020-12-10

**Authors:** Amalia Stefaniu, Lucia Pirvu, Bujor Albu, Lucia Pintilie

**Affiliations:** National Institute for Chemical-Pharmaceutical Research and Development, 112 Vitan Av., 031299 Bucharest, Romania; abujor@gmail.com (B.A.); lucia.pintilie@gmail.com (L.P.)

**Keywords:** SARS-CoV-2, benzoic acid derivatives, gallic acid, molecular docking, reactivity parameters

## Abstract

Several derivatives of benzoic acid and semisynthetic alkyl gallates were investigated by an in silico approach to evaluate their potential antiviral activity against SARS-CoV-2 main protease. Molecular docking studies were used to predict their binding affinity and interactions with amino acids residues from the active binding site of SARS-CoV-2 main protease, compared to boceprevir. Deep structural insights and quantum chemical reactivity analysis according to Koopmans’ theorem, as a result of density functional theory (DFT) computations, are reported. Additionally, drug-likeness assessment in terms of Lipinski’s and Weber’s rules for pharmaceutical candidates, is provided. The outcomes of docking and key molecular descriptors and properties were forward analyzed by the statistical approach of principal component analysis (PCA) to identify the degree of their correlation. The obtained results suggest two promising candidates for future drug development to fight against the coronavirus infection.

## 1. Introduction

Severe acute respiratory syndrome coronavirus 2 is an international health matter. Previously unheard research efforts to discover specific treatments are in progress worldwide. Virtual screening of existing compound databases against three protein targets (main protease, RNA dependent RNA polymerase and spike protein) to inhibit coronavirus replication is one of the actual approaches that allows researchers to quickly select best drug candidates for further in vitro assays [1,2,3,4,5].

Boceprevir is a direct-acting antiviral agent (DAA), acting as an inhibitor of NS3/4A, a serine protease enzyme encoded by hepatitis C virus (HCV) [6]. The serine protease enzyme plays a vital role in the replication and cleavage of viral proteins. We found boceprevir as a cocrystallized ligand in a complex with a protein named 3C-like proteinase, the main protease found in coronaviruses, characterized by X-ray diffraction, introduced in the protein data bank with the entry ID: 6WNP [7], at 1.45 Å resolution. The main protease in coronaviruses contains a cysteine-histidine dyad able to achieve catalytic cleavage of the coronavirus polyprotein [8]. Cysteine acts as a nucleophile due to its free electron pair on the sulfur atom, donated to form intramolecular bonds and histidine, respectively, that acts as a general base by its imidazole heterocycle [9]. The main protease of SARS-CoV-2 consists of three domains and a characteristic CYS145-HIS41 dyad in the active site [10].

Due to their low toxicity and antioxidant activity, phenolic acids and flavonoids appear as the most feasible and secure natural antiviral compounds. The potential antiviral activity of vegetal polyphenols is generally based on their capacity to alter virus replication and functional protein synthesis [11]. Another aspect to be mentioned is the capacity of volatile oils, saponins and triterpenic acids, to act as effective solvents and detergents, therefore to solubilize and destroy the lipid layer of the enveloped viruses. Regarding the general chemical aspects associated with antiviral activity, the presence of phenyl ring(s), vinyl and carboxyl moieties and ester, hydroxyl and methoxy groups appear to be the basis of plant phenolics antiviral efficacy [11,12,13]. For example, it is considered that phenolics with free hydroxyl groups interfere with viral adsorption and further cell penetration [11], which is sustained by the fact that high polar phenolics create a protective coating on the cell’s surface. On the other hand, the plant compounds’ bioavailability allows them to reach the circulation and be able to manifest antiviral activity. Accordingly, clinical studies have revealed that gallic acid, catechins, flavones and quercetin glucosides have the best bioavailability in humans [14]; also, data suggests that anthocyanins are fully absorbed in humans [15]. The most potent antiviral compounds proved to interfere with virus replication and/or viral essential protein synthesis are some of the most common vegetal polyphenols, namely quercetin, kaempferol and apigenin, ellagic, rosmarinic and gallic acids, catechin and epicatechin, and various alkyl gallates [11]. Similary, chrysin, acacetin and apigenin inhibited viral transcription of the human rhinovirus 14 [16], while proanthocyanidins from *Myrothamnusf labellifolia* have proved antiviral activity against herpes simplex virus type-1 by viral adsorption and cell penetration inhibition [17].

Furthermore, molecular docking studies in the last two years, made on dozens of natural and synthesized antiviral compounds, associate their antiviral activity with the capacity of their active groups (phenyl groups and phenyl moieties such as hydroxyl, carbonyl, amino, azo, nitrile, sulfonyl) to bind the active groups of several amino acids found in the active site of SARS-CoV-2 main protease. GLU166, HIS41, ASN142, GLY143, HIS163 and THR 190, are on the top of the most frequent amino acids bound [10,18,19,20].

Recent virtual screening of 22 U.S.-FDA approved antiviral drugs in the parallel with 24 natural plant-based molecules, indicated theaflavin digallate (from green tea) with the best docking score (−10.574), a result obtained using the Glide (Schrödinger) module on the COVID-19 main protease (structure 6LU7) [20]. It must be noted that excepting HIS41, which bound Osp^2^ from the carboxyl moiety of gallic acid, all other interactions occurred at hydroxyl groups from the flavan ring. Therefore, it is confirmed that the high degree of hydroxylation is associated with a good antiviral effect.

Another aspect to be mentioned is the conclusion of a computational survey to a drug repurposing study claiming that five neutral antiviral drugs have inhibitory activities against SARS-CoV-2 main protease [21]. As it is well known, the logP value describes lipophilicity for neutral compounds. Therefore, the compounds with an octanol-water partition coefficient (logP) value between one to three have good passive absorption across lipid membranes (bioavailability), while those with a logP value greater than three or less than one usually have lesser bioavailability [22]. In this context, further calculations can be made in connection with the formulation of antiviral products to obtain targeted bioavailability in the human intestine.

Gallic acid, one of the most abundant phenolic acids in plants, has various health benefits including anti-inflammatory, antioxidant and antimicrobial activities [23] and is viewed as the lead compound with promising pharmacological properties to design and develop new drugs [24]. Regarding semisynthetic derivatives of gallic acid, the alkyl gallates with antimicrobial activity were reported too [25,26]. Their biological activity is associated with the length with their alkyl side chain, which affects membrane binding capability [25]. Generally, membrane binding of the alkyl gallates increases with increasing alkyl chain length and is correlated with their antiviral activity.

Starting from such intriguing premises, in molecular docking approach we conducted in silico screening on several benzoic acid derivatives and on a homologue series of alkyl gallates, starting from the lead compound gallic acid, against SARS-CoV-2 main protease.

## 2. Results and Discussion

We selected the following compounds: benzoic acid, 4-aminobenzoic acid, 4-hydroxybenzoic acid, 3,4,5-trimethoxybenzoic acid (eudesmic acid), 3,4-dihydroxybenzoic acid (protocatehuic acid), 2,5-dihydroxybenzoic acid (gentisic acid), 4-hydroxy-3-methoxybenzaldehyde (vanillin), 4-hydroxy-3,5-dimethoxybenzoic acid (syringic acid), 4,5-dihydroxy-3-oxocyclohex-1-ene-1-carboxylic acid, epicatechin, 3,4,5-trihydroxybenzoic acid (gallic acid), methyl 3,4,5-trihydroxybenzoate, ethyl 3,4,5-trihydroxybenzoate, propyl 3,4,5-trihydroxybenzoate, isopropyl 3,4,5-trihydroxybenzoate, butyl 3,4,5-trihydroxybenzoate, isobutyl 3,4,5-trihydroxybenzoate, pentyl 3,4,5-trihydroxybenzoate, isopentyl 3,4,5-trihydroxybenzoate, octyl 3,4,5-trihydroxybenzoate. Their 3D optimized structures obtained with the Spartan program, along with the atomic numbering scheme, arbitrary chosen by the software, are given in Figure 1.

### 2.1. Results of Molecular Docking Simulations

Intermolecular interactions occurring in boceprevir, benzoic acid and alkyl gallate derivatives in complex with the 6WNP protein fragment were identified. The lengths of hydrogen-bonding interactions were measured. The results are given in terms of docking score function and RMSD (root mean square deviation).

To validate the molecular docking protocol, boceprevir was initially docked into the crystal structure of the main protease fragment and its interactions with the target 3C-like proteinase were analyzed. As can be seen Figure 2a, the native ligand forms eight hydrogen bonding interactions with residues CYS145, SER144, GLY143, HIS41, HIS164, GLU166, GLU166 and GLU166. In Figure 2b, the superposition of the binding pose of boceprevir, obtained by redocking, is displayed. As illustrated in Figure 2c, all docked ligands were found to have similar binding poses to the native ligand, thus validating the chosen docking approach.

Figure 3 reveals the obtained docking scores for the cocrystallized and selected ligands. Boceprevir exhibits the greatest score (−63.95), due to its numerous interactions with the amino acid residues from the protein’s active binding site, i.e., eight hydrogen bonding interactions with nitrogen or oxygen atoms of amino acids residues N sp^2^ CYS145 (2.900 Å), N sp^2^ SER144 (3.053 Å), N sp^2^ GLY143 (2.783 Å), N sp^2^ HIS41 (2.604 Å), O sp^2^ HIS164 (3.103 Å), N sp^2^ GLU166 (3.118 Å), O sp^2^ GLU166 (2.908 Å and O sp^2^ GLU166 (3.229 Å). Remarkable is the docking score for octyl-gallate (−60.22, RMSD 1.12), close to boceprevir, and for epicatechin (−49.57). Benzoic acid derivatives, in the first half of the graph exhibit moderate scores ranging between −29.59 (benzoic acid) and −37.25 (syringic acid). We observed an increasing trend of score with the increasing number of hydroxyl groups. For instance, gallic acid docking simulations resulted in a −38.31 score. Regarding the gallates, the increasing length of alkyl chain led to docking scores in the order gallic acid < methyl gallate < ethyl gallate < isopropyl gallate < propyl gallate < butyl gallate < isobutyl gallate < isopentyl gallate < pentyl gallate < octyl gallate. The ramification of lateral side chain led to a slight decrease of binding affinity, noticeable with an increasing number of -CH_2_ (e.g., isopentyl: −46.17 versus pentyl gallate: −48.77).

In Appendix A available online are listed the obtained values for docking score and RMSD (root mean square deviation), the amino acids group interactions and type and length of hydrogen-bonding interactions formed by each ligand in complex with SARS-CoV-2 main protease.

Figure 4 illustrates the intramolecular interactions by H-bonding (a), and amino acid group interactions occurring in the complex formed by octyl gallate and the 6WNP main protease fragment.

Amino acid residues CYS145 and SER144 interacted by hydrogen-bonding both with boceprevir and with octyl-gallate (see Figure 4a); similar interactions resulted in similar docking results. Propyl gallate and pentyl gallate revealed the same ten intramolecular interactions with N sp^2^ GLU166, N sp^2^ HIS163, O sp^3^ SER144, O sp^3^ SER144, O sp^2^ LEU141, N sp^2^ SER144, N sp^2^ GLY143, N sp^2^ CYS145, N sp^2^ GLY143 and O sp^2^ ASN142. Their obtained docking scores were −42.13 and −48.77, respectively; lower than for octyl-gallate but greater than the results for gallic acid (−38.31). These observations are in good agreement with experimental findings of Takai E. et al., 2019 [25], anticipating increasing antiviral effects of alkyl gallates with increasing alkyl chain length, except for cetyl and stearyl gallate (which are not included in this study), a fact experimentally validated by fluorescence analysis of the binding of alkyl gallates to phospholipid membranes. Beyond a certain alkyl chain length (8–11), a reduction of antibacterial and antiviral activities of the alkyl gallates was observed, probably due to a self-association process [25,26,27]. This was the reason for choosing to break off the gallates screening at octyl. Increasing docking results were observed with increasing length of the *n*-alkyl side chain. The good docking result for octyl gallate recommends it as good alternative for developing new therapeutic antiviral agents.

### 2.2. Results of Oral Bioavailability Evaluation

In Table 1 are listed key molecular descriptors and properties to evaluate the oral bioavailability [28,29] and Veber’s [30] rules, where: MW is the molecular weight that should be less than 500 Daltons, HBD is the number of hydrogen bond donors (recommended to be lower than 5), HBD is the number of hydrogen bond acceptors with acceptable values less than 10 and log P is the water-octanol partition coefficient, that should be less than 5. Veber D.F. et al., 2002 [30] imposed additional restrictions related with the molecular descriptor PSA (polar surface area), namely, no larger than 140 Å^2^ and with a maximum 10 rotatable bonds for good oral bioavailability.

Therefore, the boceprevir antiviral exhibited two violations of Lipinski’s criteria, namely molecular mass (521.69 g mol^−1^) and six hydrogen bond donors. The structure of boceprevir presents the maximum allowed number of flexible bonds (10) and maximum hydrogen bond acceptors (10). Although it had these exceptions, the docking score was the highest among the investigated ligands, suggesting strong interactions and stability of the complex formed with the SARS-CoV-2 main protease. The results indicated that all proposed ligands met the restrictive criteria for good oral bioavailability. Increased hydrophilicity was observed for all compounds due to their hydroxyl, carboxyl and/or methoxy groups on their skeleton. These functional groups offer good absorption and permeation properties. Thus, by means of NH/OH/N/O groups, hydrophilic interactions were favored and further reflected in good and high docking scores. Concerning logP values, there were observed positive values for benzoic acid (0.79), 2,5-dihydroxybenzoic acid (0.81) and 0.72 for octyl 3,4,5-trihydroxy benzoate. A combination of molecular factors and properties, mainly due to the increased hydrophobicity of the lateral *n*-octyl chain of octyl gallate, was also reflected by its best docking score function, indicating this compound as the best antiviral candidate among all screened compounds in the study.

### 2.3. Results of Quantum Reactivty Analysis

Frontier molecular orbitals, the highest occupied molecular orbital (HOMO) and the lowest unoccupied molecular orbital (LUMO) localization and energy levels for octyl-gallate, are illustrated in Figure 5.

The resulting band gap (ΔE) can provide useful information on the chemical reactivity and kinetic stability of each ligand. The same values for energy gap were given for alkyl gallates, starting at ethyl to octyl-gallate, suggesting the same stability. Slight differences were found for the values of ionization potential (I = −E_HOMO_) and electron affinity (A = −E_LUMO_), according Koopmans’ theorem [31]. The theorem allows estimation of quantum global reactivity parameters, starting from calculated energies of frontier molecular orbitals, and describes the molecules in terms of chemical hardness (η), global softness (σ), ionization (I), electron affinity (A), electronegativity (χ) and electrophyliciy index (ω) [32,33]. Obtained quantum reactivity parameters for all investigated ligands are given in Appendix A.

The global reactivity parameters analysis provides deep structural insights, a holistic characterization for revealing the properties of interest leading to strong binding affinity related to the protease target. The quantum reactivity parameters are related to relative nucleophilicity and electrophilicity. Ionization potential (I), refers to the energy needed to remove an electron from a molecule, and electron affinity (A) measures the ability of a molecule to accept electrons and form anions species [34,35]. Such parameters are useful to estimate further chemical reactivity behavior. Some of the investigated molecules can also be seen as lead compounds for a new series of (semi)synthetic molecules. Therefore the data on their reactivity are useful.

### 2.4. Results of Principal Component Analysis (PCA)

Principal component analysis (PCA) is a statistical tool for the identification of linear combinations of the variables which account for certain proportions of the variance of a set of variables. The selection is based on the eigenvalues of the dispersion matrix of the variables. The principal components are associated with decreasing eigenvalues and, therefore, share the amount of the variance. Typically, the first few principal components account for virtually all the variance. PCA also represents the pattern of similarity of the observations and the variables by displaying them as points in maps [36,37,38,39]. PCA analysis of all properties was calculated with Spartan software, along with docking scores, and data are listed in Appendix A.

PCA analysis was employed to find the degree of correlation between molecular descriptors and properties and their involvement in the resulting docking score.

The PCA correlation matrix (Appendix A) revealed fairly good correlations between area and mass (r = 0.95), area and ovality (r = 0.96), docking score and polarizability (r = 0.97), volume and area (r = 0.97), volume and ovality (r = 0.98) and a moderate correlation (r = 0.55–0.66) between the dipole moment and docking score, mass, area, volume, and ovality, and between PSA and polarizability and mass, respectively.

Table 2 and Figure 6 are related to the eigenvalues which reflect the quality of the projection from the N–dimensional initial (*n* = 6) to a lower number of dimensions.

From Table 2 it can be seen that the first eigenvalue equals 6.529 and represents 65.29 % of the total variability. Each eigenvalue corresponds to a factor, and each factor to a one dimension. A factor is a linear combination of the initial variables and all the factors are uncorrelated (r = 0). The eigenvalues and the corresponding factors are sorted by descending order of how much of the initial variability they represent (converted to %). Therefore, the first two factors allow us to represent 70.69% of the initial variability of the data.

## 3. Methods

### 3.1. Methods for Molecular Docking Simulations

The docking simulation was carried out using CLC Drug, Discovery Work Bench (QIAGEN, Aarhus, Denmark,). SARS-CoV-2 main protease bound to boceprevir at 1.45 Å (PDB ID: 6WNP) [7], which was imported from the Protein Data Bank. Ligands were constructed using the Spartan’16 program [40,41] and their geometries were optimized to obtain the lowest energy conformers. The viral protein fragment contains three binding pockets: 48.13 Å^3^, 40.45 Å^3^, 36.62 Å^3^. Redocking of the cocrystallized ligand (boceprevir) was realized to validate the docking protocol. The amino acid residues forming the binding site were protonated and water molecules were removed.

### 3.2. Methods for Energy Minimization and Computation of Molecular Properties

The examined ligands were first generated in 3D by importing their corresponding files from the Pubchem database (National Center for Biotechnology Information, U.S. National Library of Medicine Rockville Pike, Bethesda, MD, USA, https://pubchem.ncbi.nlm.nih.gov) or directly constructed (long chain alkyl gallates) in the Spartan’18 program [40,41]. Their geometry was optimized in a multistep procedure by molecular mechanics force fields, developed at Merck Pharmaceuticals (Merck Research Labs, Kenilworth, NJ, USA), to realize energy minimization [42]. Molecular descriptor and properties were calculated using the DFT, B3LYP algorithm [43] and the 6-31G (d, p) polarization basis set [40]. Calculations were performed for equilibrium geometry at ground state in gas for neutral molecules. No solvent corrections were done. 

### 3.3. Methods for Principal Component Analysis (PCA)

The predicted ligand data were processed to compute principal component analysis using the free Excel add-in Real Statistics Resource Pack software (Release 7.2, www.real-statistics.com), Copyright (2013–2020) Charles Zaiontz (www.real-statistics.com).

## 4. Conclusions

In this work we analyzed and predicted the antiprotease activity of natural derivatives of benzoic and semisynthetic alkyl gallate acids on SARS-CoV-2 main protease. The investigation was corroborated with drugability and quantum reactivity evaluations. The docking results of the two studied series (benzoic acid series versus gallic acid series) suggested 2,5-dihydroxybenzoic acid (gentisic acid) and octyl gallate as the best potential candidates among the investigated structures. The two compounds had similar logP values (0.81 and 0.72). Octyl gallate had the best docking score (−60.22), but decreased dipole moment (1.31). 2,5-dihydroxybenzoic acid (gentisic acid) had a lower score (−33.84) but an increased dipole moment (4.62) which means higher polarity and also higher reactivity. As is known, the dipole moment represents a measure of net molecular polarity and, therefore, the larger the difference in electronegativities of bonded atoms the larger the reactivity of the molecule. Accordingly, a combination of the two compounds can be considered. On the other hand, if the benzoic acid series indicated that the position of hydroxyl groups on the ring was more important than the ester, hydroxyl and methoxy groups’ number or type, the galllic acid series clearly indicated similar results for all studied compounds, apart from octyl gallate which was proved to be the best potential candidate among investigated structures, exhibiting antiviral activity against the SARS-CoV-2 protease and, therefore, could be studied further for drug development.

## Figures and Tables

**Figure 1 molecules-25-05828-f001:**
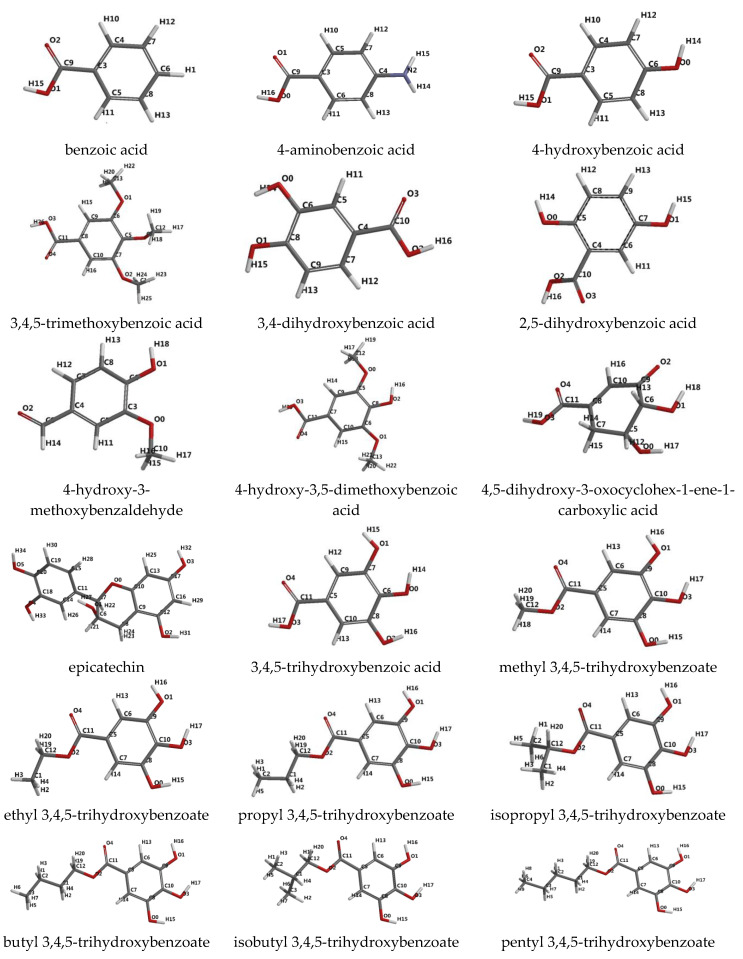
3D optimized structures of investigated ligands and their numbering atomic labels.

**Figure 2 molecules-25-05828-f002:**
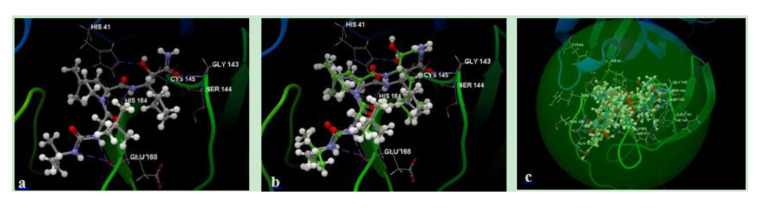
Hydrogen bonding interactions of the native ligand (boceprevir); (**a**) superposition of the native ligand; (**b**) superposition of all docked ligands in the active binding site of 3C-like proteinase (6WNP) (**c**).

**Figure 3 molecules-25-05828-f003:**
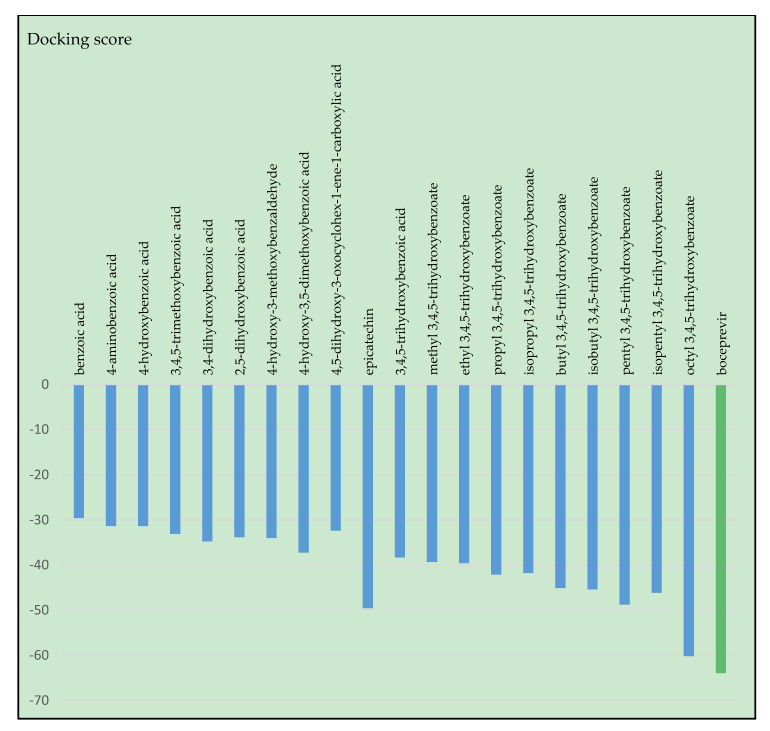
Docking scores for investigated ligands against SARS-CoV-2 main protease (6WNP).

**Figure 4 molecules-25-05828-f004:**
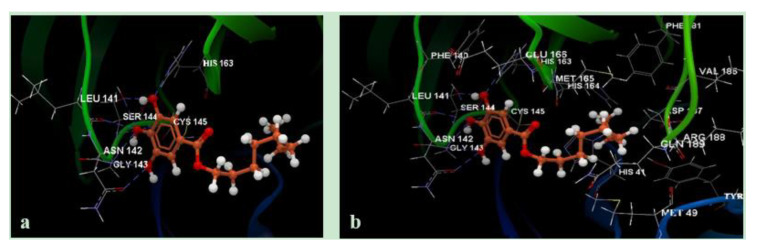
Hydrogen bonding interactions of octyl-gallate (**a**) and interactions with amino acid residues from the active binding site of 6WNP protein fragment (**b**).

**Figure 5 molecules-25-05828-f005:**
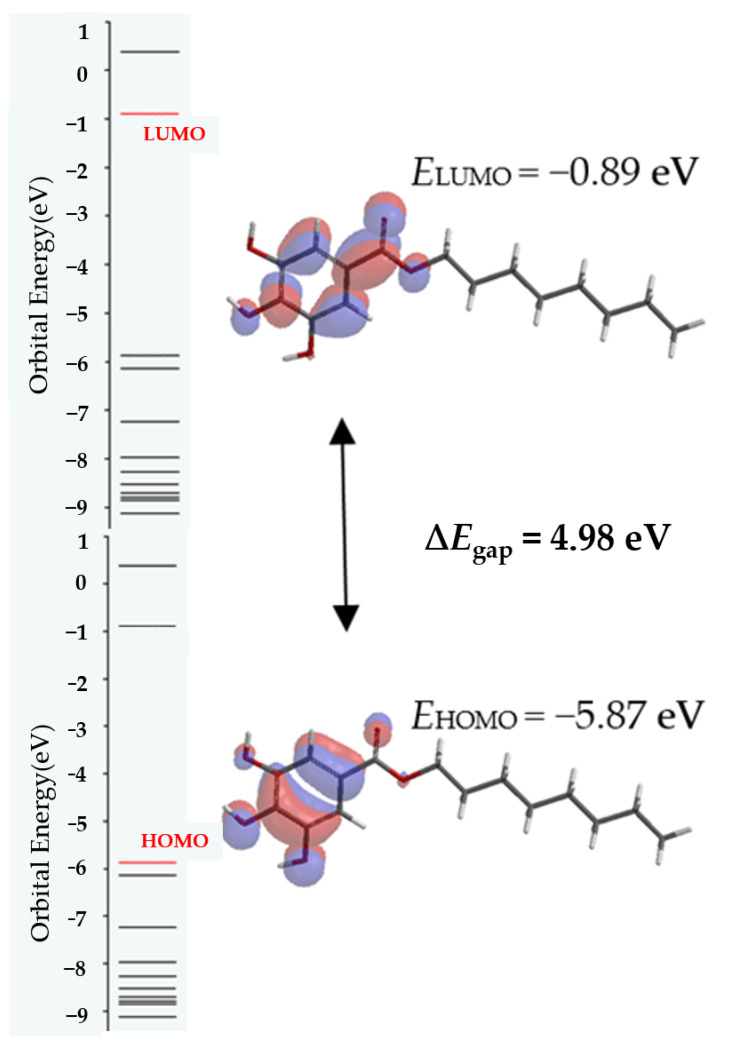
Highest occupied molecular orbital (HOMO) and lowest unoccupied molecular orbital (LUMO) molecular frontier orbitals and their energy gap for octyl-gallate.4

**Figure 6 molecules-25-05828-f006:**
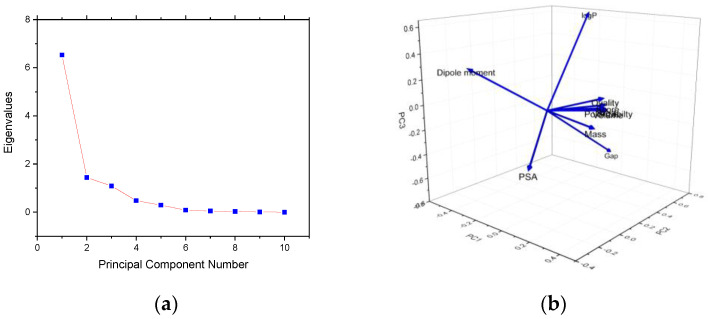
Results of PCA analysis: scree plot of the eigenvalues and cumulative variability versus the F1-F6 components (**a**) and PC1 PC2 PC3 3D Loading plot (**b**). The scree plot (Figure 6a) is a useful visual aid for determining the number of the principal components, which depends on the elbow point at which the remaining eigenvalues are relatively small and all about the same size. This point is not so evident in the scree plot, but we may say that our elbow point is the third point. In conclusion Plot 4 and the Table 2 indicate that the first three PCs are sufficient to explain most of the variance (more than 90.56%) of the data set without overfitting the model. Detailed information about the best principal component locations of the extracted eigenvectors PC1-PC2-PC3 3D loading plots are also presented (Figure 6b). It is worth mentioning that the dipole moment is located in the PC2 × PC3 space, PSA in the PC1 × PC2 space and the other variables in the PC1 × PC3 space.

**Table 1 molecules-25-05828-t001:** Lipinski and Veber’s parameters for drugability assessment.

Ligand	MW	PSA	HBD	HBA	LogP	rb	LV
Benzoic acid	122.123	33.690	1	2	0.79	1	0
4-Aminobenzoic acid	137.138	58.471	3	3	−0.93	1	0
4-Hydroxybenzoic acid	138.122	53.444	2	3	−0.29	1	0
3,4,5-Trimethoxybenzoic acid	212.201	53.223	1	5	−2.14	4	0
3,4-Dihydroxybenzoic acid	154.121	71.217	3	4	−1.37	1	0
2,5-Dihydroxybenzoic acid	154.121	71.262	3	4	0.81	1	0
4-Hydroxy-3-methoxybenzaldehyde	152.149	41.012	1	3	−1.53	2	0
4-Hydroxy-3,5-dimethoxybenzoic acid	198.174	64.706	2	5	−2.24	3	0
4,5-Dihydroxy-3-oxocyclohex-1-ene-1-carboxylic acid	172.136	83.671	3	5	−0.92	1	0
Epicatechin	290.271	101.294	5	6	−3.72	1	0
3,4,5-Trihydroxybenzoic acid	170.12	89.408	4	5	−2.46	1	0
Methyl 3,4,5-trihydroxybenzoate	184.147	75.752	3	5	−2.19	2	0
Ethyl 3,4,5-trihydroxybenzoate	198.174	75.425	3	5	−1.86	3	0
Propyl 3,4,5-trihydroxybenzoate	212.201	75.433	3	5	−1.37	4	0
*i*-Propyl 3,4,5-trihydroxybenzoate	212.201	75.068	3	5	−1.54	3	0
Butyl 3,4,5-trihydroxybenzoate	226.228	75.433	3	5	−0.95	5	0
*i*-Butyl 3,4,5-trihydroxybenzoate	226.228	75.149	3	5	−0.97	4	0
Pentyl 3,4,5-trihydroxybenzoate	240.255	75.426	3	5	−0.54	6	0
*i*-Pentyl 3,4,5-trihydroxybenzoate	240.255	75.415	3	5	−0.62	5	0
Octyl 3,4,5-trihydroxybenzoate	282.336	75.390	3	5	0.72	9	0

MW—molecular weight (g mol^−1^); PSA—polar surface area (Å^2^) HBD—hydrogen bond donor; HBA—hydrogen bond acceptor; rb—rotatable bonds count; LV—Lipinski’s violations.

**Table 2 molecules-25-05828-t002:** Eigenvalues from the Principal Component Analysis (PCA) analysis.

	F1	F2	F3	F4	F5	F6
Eigenvalue	6.529	1.440	1.087	0.480	0.291	0.084
Variability, %	65.29%	14.40%	10.87%	4.80%	2.91%	0.84%
Cumulative, %	65.29%	79.69%	90.56%	95.36%	98.27%	99.11%

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
