# Peer review of "Molecular Docking Study on Several Benzoic Acid Derivatives against SARS-CoV-2"

_molecules, 2020, doi:10.3390/molecules25245828_

Round 1

Reviewer 1 Report

The manuscript of “molecules-1020868” by Stefaniu et. al. mainly presents a molecular docking study of ~19 molecules against SARS-Cov-2 main protease. Additional methods, such as DFT calculations, druglikness assessment, etc. are used to further analyze the molecules selected in this manuscript. In general, this study provides computational guidance of potential inhibitor design for SARS-CoV-2 main protease. However, there are several issues in this manuscript that should be addressed before publication.

  1. There are too many grammar/spelling issues in the manuscript to list specifically. The low quality of the presentation may cause great trouble for readers to understand the methods and results presented in this study.
  2. I am a bit confused by the DFT calculation used in this study to assess the reactivity of the docked molecules. These molecules should be non-covalent binders so why do the authors need to study the reactivity of these molecules?
  3. Also, what is the purpose of using the principal components analysis? Instead of visualizing the results, I didn’t see how the authors refined the final conclusion using the results from PCA.

Author Response

Reviewer 1

Comments and Suggestions for Authors

The manuscript of “molecules-1020868” by Stefaniu et. al. mainly presents a molecular docking study of ~19 molecules against SARS-Cov-2 main protease. Additional methods, such as DFT calculations, druglikness assessment, etc. are used to further analyze the molecules selected in this manuscript. In general, this study provides computational guidance of potential inhibitor design for SARS-CoV-2 main protease. However, there are several issues in this manuscript that should be addressed before publication.

Q1: There are too many grammar/spelling issues in the manuscript to list specifically. The low quality of the presentation may cause great trouble for readers to understand the methods and results presented in this study.

A1: The grammar/spelling issues have been corrected.

Q2: I am a bit confused by the DFT calculation used in this study to assess the reactivity of the docked molecules. These molecules should be non-covalent binders so why do the authors need to study the reactivity of these molecules?

A2: The study aimed to provide deep structural insights, a holistic characterization for revealing the properties of interest leading to strong binding affinity related to protease target. The quantum reactivity parameters are related with the relative nucleophilicity and electrophilicity. Ionization potential (I), refers to the energy needed to remove an electron from a molecule; electron affinity (A), measures the ability of a molecule to accept electrons and form anions species. Such parameters are useful for estimate further chemical reactivity behavior. Some of the investigated molecules can also be seen as leads compounds for new series of (semi)synthetic molecules, therefore the data on their reactivity are useful.

            In terms of interactions ligand-protein, hydrogen bonding type are expected and depicted.

               A paragraph resuming the answer of your question, in order to make the quantum reactivity analysis more explicit, accompanied by citing references, was added at Section 3.3. Results of quantum reactivity analysis.

Q3: Also, what is the purpose of using the principal components analysis? of visualizing the results, I didn’t see how the authors refined the final conclusion using the results from PCA.

A3: The purpose of using the principal components analysis was to find the degree of correlation between molecular descriptors and properties and their involvement in the resulting docking score. Good correlations are  observed between area and mass (r = 0.95), area and ovality (r = 0.96), docking score and polarizability (r = 0.97), volume and area (r = 0.97), volume and ovality (r = 0.98), and a moderate correlation (r = 0.55 – 0.66), between the dipole moment and docking score, mass, area, volume, ovality respectively, and PSA and polarizability and mass, respectively (as can be see form Table S4 from Supllementary material.

Thank you for your valuable observations and suggestions!

Reviewer 2 Report

Paper title: Molecular docking study on several benzoic acid derivatives against SAR-COV-2.

The aim of the study is clear and authors provided adequate information on how they conclude to their results. The references are relevant and generally recent and include appropriate studies, but currently, there are several manuscripts discussing molecular docking studies against sars-cov-2 related proteins and the authors should include them to increase the size of their introduction as well. It is clear what is already known about the topic, but a paragraph should be included in introduction to explain more the potential mechanisms of action of the studied molecules from the literature. The research question is clearly outlined and almost justified (the paragraph should solve this). The process is valid and the variables are defined appropriately, but in order to improve further the validity of their methods the authors should provide deeper explanation about the validation of their method in order for their results to be reproducible by other researchers and the manuscript be cited by others in the future.

The data are presented in an appropriate way, except of table 1 in page 7 which should be changed to table 2, and by mistake is labelled as table 1 again. The authors used appropriate units and titles are clearly represented. The text to the data adds to the results and the practical results are clear as well. There also discussed from multiple angles and the final conclusions answer the aim of the study as it should. Some conclusions are also supported by references and generally speaking the results are opportunities for further research. Despite that, I believe that this article provides information as acronyms and abbreviations that is really valuable, thus an acronym and abbreviation section should include in order for the paper to be completely understood by a general reader. I also believe that authors should merge figures 4 and 5 to one figure. Further to this, figure S1 from the supplementary should be moved to the main manuscript and a figure including the studied protein with its binding site should be included. Finally, in conclusions section in line 2, a space should be placed between sars-cov2 and main protease.

In my opinion the article should be published with minor improvements. The English is very good and understandable and appropriate for a scientific manuscript.

Best regards

Author Response

Reviewer 2

Comments and Suggestions for Authors

Q1. The aim of the study is clear and authors provided adequate information on how they conclude to their results. The references are relevant and generally recent and include appropriate studies, but currently, there are several manuscripts discussing molecular docking studies against sars-cov-2 related proteins and the authors should include them to increase the size of their introduction as well. It is clear what is already known about the topic, but a paragraph should be included in introduction to explain more the potential mechanisms of action of the studied molecules from the literature.

A1. Several additional references related with molecular docking approaches were introduced and discussed in the Introduction section.

  • Shah, B.; Modi, P.; Sagar, S. R. In silico studies on therapeutic agents for COVID-19: Drug repurposing approach. Life Sci. 2020, 252, 117652, https://doi.org/10.1016/j.lfs.2020.117652
  • Cardoso, W.B.; Mendanha, S.A. In silico studies on therapeutic agents for COVID-19: Drug repurposing approach. Mol. Struct. 2021, 1225, 129143, https://doi.org/10.1016/j.lfs.2020.117652
  • Peele, K.A.; Durthi, C.P.; Srihansa, T.; Krupanidhi, S.; Sai, A.V.; Babu, D.J.; Indira, M.; Reddy, A.R.; Venkateswarulu, T.C.; Inform. Med. Unlocked. 2020, 19, 100345, https://doi.org/10.1016/j.imu.2020.100345
  • Wang, J. Fast identification of possible drug treatment of coronavirus disease-19 (COVID-19) through computational drug repurposing study. Chem. Inf. Model. 2020, 60(6), 3277-3286, DOI: 10.1021/acs.jcim.0c00179
  • Hetal, T.; Bindesh, P.; Sneha, T. A review on techniques for oral bioavailability enhancement of drugs. Int. J. Pharm. Sci. Rev. Res. 2010, 4(3), 203-223.

Data about component domains of the main protease of SARS-CoV-2 and its active site characteristic CYS145-HIS41 dyad was discussed.  The binding way of various structures form recent literature data were exemplified. Frequently, they bind mainly by their active groups (phenyl groups and phenyl moieties such as hydroxyl, carbonyl, amino, azo, nitrile, sulfonyl), to the amino acid residues from the active site of SARS-CoV-2 main protease: GLU166, HIS41, ASN142, GLY143, HIS163 and THR 190.

Q2. The research question is clearly outlined and almost justified (the paragraph should solve this). The process is valid and the variables are defined appropriately, but in order to improve further the validity of their methods the authors should provide deeper explanation about the validation of their method in order for their results to be reproducible by other researchers and the manuscript be cited by others in the future.

A2. A paragraph including details about docking validation was introduced and figures illustrating the native ligand docking pose and validation, by superposing the results of re-docking Boceprevir and its hydrogen bonding interactions, were added. Also, the similar docking poses of investigated ligands, are presented, as follows:

To validate the molecular docking protocol, Boceprevir was initially docked into the crystal structure of the main protease fragment and its interactions with the target 3C-like proteinase were analyzed. As can be seen Figure 2a, the native ligand forms eight hydrogen bonding interactions with residues: CYS145, SER144, GLY143, HIS41, HIS164, GLU166, GLU166 and GLU166. In Figure 2b, the superposition of the binding pose of Boceprevir, obtained by re-docking, is displayed. As illustrated in Figure 2c, all docked ligands were found to have similar binding poses to the native ligand, thus validating the chosen docking approach.

Q3. The data are presented in an appropriate way, except of table 1 in page 7 which should be changed to table 2, and by mistake is labelled as table 1 again.

A3. This mistake has been corrected.

Q4. The authors used appropriate units and titles are clearly represented. The text to the data adds to the results and the practical results are clear as well. There also discussed from multiple angles and the final conclusions answer the aim of the study as it should. Some conclusions are also supported by references and generally speaking the results are opportunities for further research. Despite that, I believe that this article provides information as acronyms and abbreviations that is really valuable, thus an acronym and abbreviation section should include in order for the paper to be completely understood by a general reader.

A4. The abbreviations list has been added.

Q5. I also believe that authors should merge figures 4 and 5 to one figure. Further to this, figure S1 from the supplementary should be moved to the main manuscript and a figure including the studied protein with its binding site should be included.

A5. Figures 4 and 5 have been merged in Figure 4a and b).

Figure 4. Results of PCA analysis: scree plot of the eigenvalues and cumulative variability versus the F1-F6 components (a) and PC1 PC2 PC3 3D Loading plot (b).

Figure S1 was moved into the main manuscript (becoming Figure 1).

A figure including the active binding site of SAR-CoV-2 main protease with all fitted ligand poses has been added (Figure 2c).

Q6. Finally, in conclusions section in line 2, a space should be placed between sars-cov2 and main protease.

A6. The correction has been made.

In my opinion the article should be published with minor improvements. The English is very good and understandable and appropriate for a scientific manuscript.

Other modifications to the manuscript:

  1. The author Bujor ALBU was placed in a rectangular shape;
  2. Rearrangement and renumbering of Figures was occur, due to the introduction of a new Figure illustrating (no. 2): Hydrogen bonding interactions of the native ligand (boceprevir) (a); superposition of the native ligand (b) and superposition of all docked ligands in the active binding site of 3C-like proteinase (6WNP) (c). and to the transfer of Figure S1 (optimized structures of ligands along with their atom labels) from supplementary files to the main text of the manuscript.
  3. Figure 1. Docking scores for investigated ligands against SARS-CoV-2 main protease (6WNP) has become Figure 3. Negative values of docking scores, are resulted form the CLC docking program, were used for the histogram and not their absolute values, as previously given, in the first draft. When comparing scores, we also refer to their absolute values (changes have been made in the text: the sign ”minus” before the score value).
  4. Spelling issues were solved.
  5. All modifications made by the authors, following the reviewer’s comments and suggestions, are written in red in the original draft manuscript.

Thank you for your valuable comments and suggestions!